# Opposing actions of co-released GABA and neurotensin on the activity of preoptic neurons and on body temperature

Iustin V Tabarean*

Scintillon Institute, San Diego, United States

**\*For correspondence:**
tabarean@scintillon.org (IVT);
tabarean@scintillon.org (IVT)

**Competing interest:** The author declares that no competing interests exist.

**Abstract** Neurotensin (Nts) is a neuropeptide acting as a neuromodulator in the brain. Pharmacological studies have identified Nts as a potent hypothermic agent. The medial preoptic area, a region that plays an important role in the control of thermoregulation, contains a high density of neurotensinergic neurons and Nts receptors. The conditions in which neurotensinergic neurons play a role in thermoregulation are not known. In this study, optogenetic stimulation of preoptic Nts neurons induced a small hyperthermia. In vitro, optogenetic stimulation of preoptic Nts neurons resulted in synaptic release of GABA and net inhibition of the preoptic pituitary adenylate cyclase-activating polypeptide (Adcyap1) neurons firing activity. GABA-A receptor antagonist or genetic deletion of Slc32a1 (VGAT) in Nts neurons unmasked also an excitatory effect that was blocked by a Nts receptor 1 antagonist. Stimulation of preoptic Nts neurons lacking *Slc32a1* resulted in excitation of *Adcyap1* neurons and hypothermia. Mice lacking *Slc32a1* expression in Nts neurons presented changes in the fever response and in the responses to heat or cold exposure as well as an altered circadian rhythm of body temperature. Chemogenetic activation of all Nts neurons in the brain induced a 4–5°C hypothermia, which could be blocked by Nts receptor antagonists in the preoptic area. Chemogenetic activation of preoptic neurotensinergic projections resulted in robust excitation of preoptic *Adcyap1* neurons. Taken together, our data demonstrate that endogenously released Nts can induce potent hypothermia and that excitation of preoptic *Adcyap1* neurons is the cellular mechanism that triggers this response.

## eLife assessment

This is an **important** study to reveal local circuit mechanisms in the POA that control body temperature and also highlight how neurotransmitter GABA and neuropeptide NTS from the same neurons differentially modulate temperature. This study was carefully executed, providing **convincing** evidence for the conclusions in this paper. The findings have emphasized the importance of considering multiple diverse functions of the same neuron populations and will be of interest to neuroscientists working on central regulations of energy metabolism and temperature homeostasis.

## Introduction

Homeotherms, including mammals, maintain core body temperature (CBT) within a narrow range, an essential requirement for survival. The preoptic area of the hypothalamus plays an important role in CBT regulation by integrating peripheral thermal information and by sending efferent signals that control thermal effector organs (*Morrison and Nakamura, 2019*; *Nakamura et al., 2022a*; *Tan and Knight, 2018*). Via projections to the dorsomedial hypothalamus and rostral raphe pallidus, preoptic

neurons control thermogenesis as well as heat loss mechanisms (*Morrison, 2018*). The preoptic area also orchestrates the fever response, an important mechanism for fighting infection or inflammatory disease (*Morrison and Nakamura, 2019*; *Saper et al., 2012*). Recent studies have identified populations of preoptic neurons that integrate peripheral thermal information and control the core body temperature (CBT) and play an important role in the fever response to endotoxin (*Machado et al., 2020*; *Moffitt et al., 2018*; *Nakamura et al., 2022b*; *Tan et al., 2016*).

Neurotensin (Nts) is a 13 aminoacid peptide found in the CNS as well as in the gastrointestinal tract (*Vincent et al., 1999*). Nts-producing neurons and their projections are widely distributed in the brain, which may explain the wide variety of effects of this peptide (*Schroeder and Leinninger, 2018*). Pharmacological approaches have revealed a role of Nts in analgesia, arousal, blood pressure, feeding, reward, sleep, the stress response and thermoregulation (*Kleczkowska and Lipkowski, 2013*; *Kyriatzis et al., 2024*; *Ramirez-Virella and Leinninger, 2021*; *Torruella-Suárez and McElligott, 2020*). Recently, studies have started to unravel the role played by specific populations of neurotensinergic neurons in feeding (*Chen et al., 2022*; *Woodworth et al., 2017*) and drinking behaviors (*Kurt et al., 2022*) as well as in sleep (*Ma et al., 2019*; *Zhong et al., 2019*) and appetitive and reinforcing aspects of motivated behaviors (*McHenry et al., 2017*).

When infused intracerebroventricularly or in the preoptic area Nts induces a 4–5°C hypothermia (*Gordon et al., 2003*; *St-Gelais et al., 2006*; *Tabarean, 2020*). The effect was attributed to activation of heat loss mechanisms (*Handler et al., 1994*). The medial preoptic area (MPO) expresses high densities of both neurotensinergic neurons and neurotensin receptors (*Schroeder and Leinninger, 2018*; *Alexander and Leeman, 1998*; *Nicot et al., 1994*). A previous study has identified a population of MPO Nts neurons that express Slc32a1 and modulate sexual behavior via projections to the ventral tegmental area (*McHenry et al., 2017*). The present study has investigated the cellular characteristics of MPO Nts neurons, their influence on the activity of nearby MPO neurons and on CBT as well as the effect of Slc32a1 deletion on these actions.

## Results

### MPO$^{Nts;ChR2}$ neurons are GABAergic and their optogenetic stimulation decreases the firing rate of MPO$^{Adcyap1}$ neurons by increasing the frequency of IPSCs

*Nts*-cre mice received bilateral MPO injections of AAV5-EF1a-double floxed-hChR2(H134R)-EYFP-WPRE-HGHpA (see Methods) to express the excitatory opsin ChR2-EYFP in MPO *Nts* neurons. We refer to these mice as MPO$^{Nts;ChR2}$. Patch-clamp recordings revealed that MPO$^{Nts;ChR2}$ neurons were spontaneously active and the majority of them (16 out of 22 neurons studied) fired mostly doublets or triplets of action potentials (*Figure 1A*) while the rest fired single action potentials. The average firing rate at 36 °C was 4.29±2.45 Hz (n=22). Intrinsic properties of PO/AH neurons were investigated using injections of square current pulses delivered in whole-cell configuration. The average capacitance of the neurons tested was 17.86±2.48 pF (n=22). All neurons tested displayed a low threshold spike (LTS) upon the end of a hyperpolarizing current injection (*Figure 1B*). Injection of positive currents generated bursts of doublets or triplets of action potentials (*Figure 1B*). To verify the presence of *Nts* transcripts in MPO$^{Nts;ChR2}$ we have carried out scRT/PCR in these neurons. We could detect *Nts* transcripts in 7 out of 10 MPO$^{Nts;ChR2}$ studied (*Figure 1C*). We have also tested the expression of the neuronal markers *Slc32a1* (VGAT) and *Slc17a6* (Vglut2). *Slc32a1* was detected in 8 out of 10 *Nts*-expressing neurons (*Figure 1C*) while none of these cells expressed *Slc17a6* (*Figure 1—figure supplement 1*) suggesting that the majority of MPO$^{Nts;ChR2}$ are GABAergic.

Spot illumination of MPO$^{Nts;ChR2}$ neurons instantly depolarized them and increased their firing rate in all neurons tested (not shown). Recordings from non-labeled MPO neurons revealed that spot illumination of nearby MPO$^{Nts;ChR2}$ neurons resulted in a robust decrease in their firing rate (9 out of 30 neurons studied) or no effect in the others (21 out of 30 neurons studied). *Figure 2A* illustrates the decrease in firing rate of a MPO neuron (from 2.2 Hz to 0.9 Hz) induced by optogenetic stimulation of a nearby MPO$^{Nts;ChR2}$ neuron. The effect was associated with a robust increase in the frequency of IPSPs from 0.5 Hz to 21.8 Hz (*Figure 2A*, inset). Overall, optogenetic stimulation of nearby MPO$^{Nts;ChR2}$ neurons reduced the firing rate of the recorded MPO neurons from 5.10±1.32 Hz to 1.39±0.36 Hz (n=9). The control value for the firing rate was calculated as the average value during the 5 min period

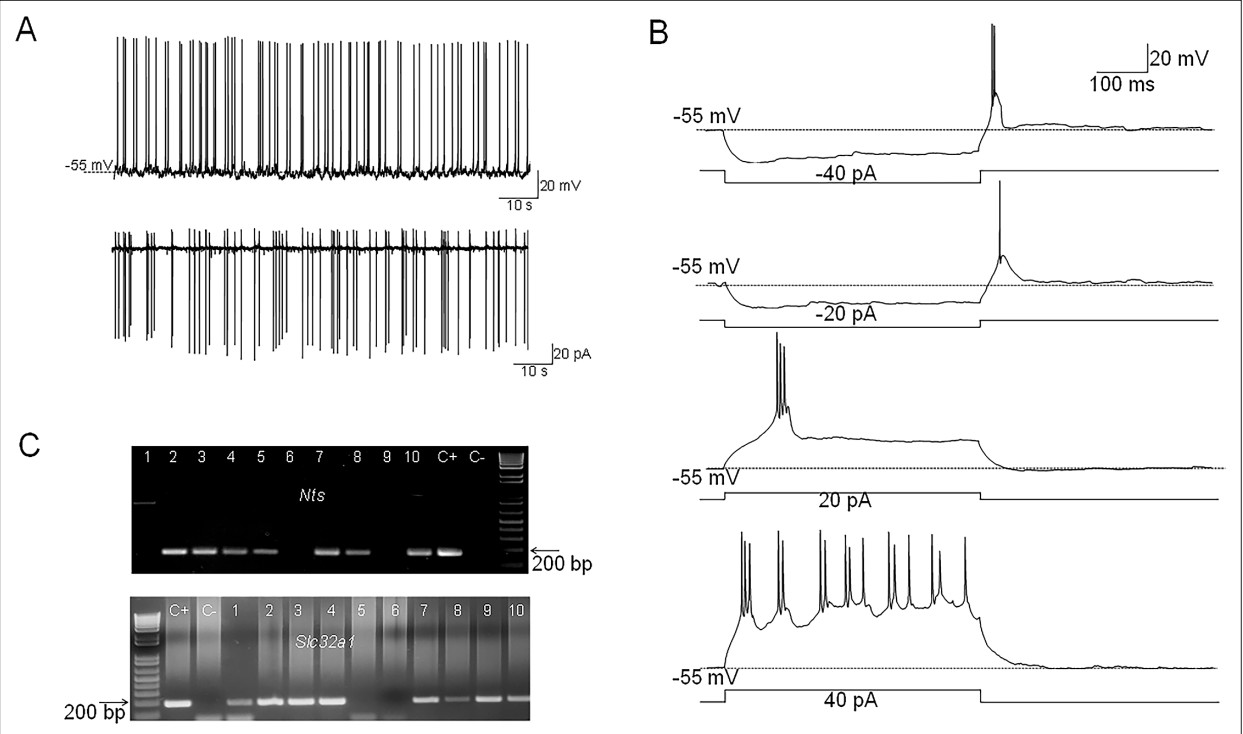

**Figure 1.** Electrophysiological characteristics of MPO$^{Nts;ChR2}$ neurons. (**A**) Representative example of spontaneous firing activity of MPO$^{Nts;ChR2}$ neurons recorded in whole-cell (up) or cell-attached configuration (down). (**B**) Membrane potential responses to hyperpolarizing current steps of –40 and –20 pA reveal the presence of a low threshold spike (LTS) upon depolarization to the resting membrane potential. Depolarizing current injections of 20 and 40 pA (right) elicit burst firing activity. The neuron fires 2–3 action potentials during each burst. (**C**) *Nts* (up) and *Slc32a1*(*VGAT*) (down) transcripts are present in MPO$^{Nts;ChR2}$ neurons. Representative results from 10 MPO$^{Nts;ChR2}$ neurons. The expected sizes of the PCR product are 149 and 137 base pairs, respectively. Negative (−) control was amplified from a harvested cell without reverse-transcription, and positive control (+) was amplified using 1 ng of hypothalamic mRNA. *Nts* transcripts were detected in 7 out of 10 neurons while *Slc32a1* was detected in 8 neurons. Six neurons expressed both transcripts.

The online version of this article includes the following source data and figure supplement(s) for figure 1:

**Source data 1.** Spontaneous firing rates and cell capacitance values for the recorded cells.

**Source data 2.** Uncropped images of the gels presented in *Figure 1*.

**Figure supplement 1.** Lack of Slc17a6 (Vglut2) expression in MPO$^{Nts}$ neurons.

preceding the optogenetic stimulation. Upon ending the photostimulation a transient increase in firing rate was observed (*Figure 2A*). We therefore studied the effect of photostimulation during incubation with the GABA-A receptor antagonist gabazine (5 µM). In the presence of the antagonist, the same neurons displayed the opposite effect, an increase in firing rate in response to photostimulation. Their firing rate was 7.89±2.61 Hz (n=9) in the presence of gabazine and increased to 9.57±2.83 Hz (n=9) during photostimulation (*Figure 2A and B*). Since we have previously found that exogenous Nts applied locally increased the firing activity of MPO neurons by activating NtsR1 (*Tabarean, 2020*) we have tested the effect of photostimulation in the presence of both gabazine (5 µM) and the NtsR1 antagonist SR48692 (100 nM). The antagonist blocked the effect of photostimulation indicating that the increase in firing rate depended on activation of NtsR1. MPO thermoregulatory neurons are a subgroup of the Adcyap1-expressing MPO population (*Moffitt et al., 2018*; *Tan et al., 2016*). We have studied whether the MPO neurons excited by photostimulation (in the presence of gabazine) express *Adcyap1* transcripts. We have detected *Adcyap1* transcripts in 9 out of 12 neurons tested. *Figure 2C* depicts an example of scRT/PCR analysis of a batch of six recorded neurons.

To further characterize the mechanisms involved in the modulation of the firing rates of MPO$^{Adcyap1}$ neurons in response to optogenetic stimulation of MPO$^{Nts;ChR2}$, we carried out additional experiments in voltage-clamp mode. In the presence of the glutamate receptor blockers CNQX (10 µM) and AP-5 (50 µM), as well as of the NtsR1antagonist SR48692 (100 nM), photostimulation MPO$^{Nts;ChR2}$ neurons

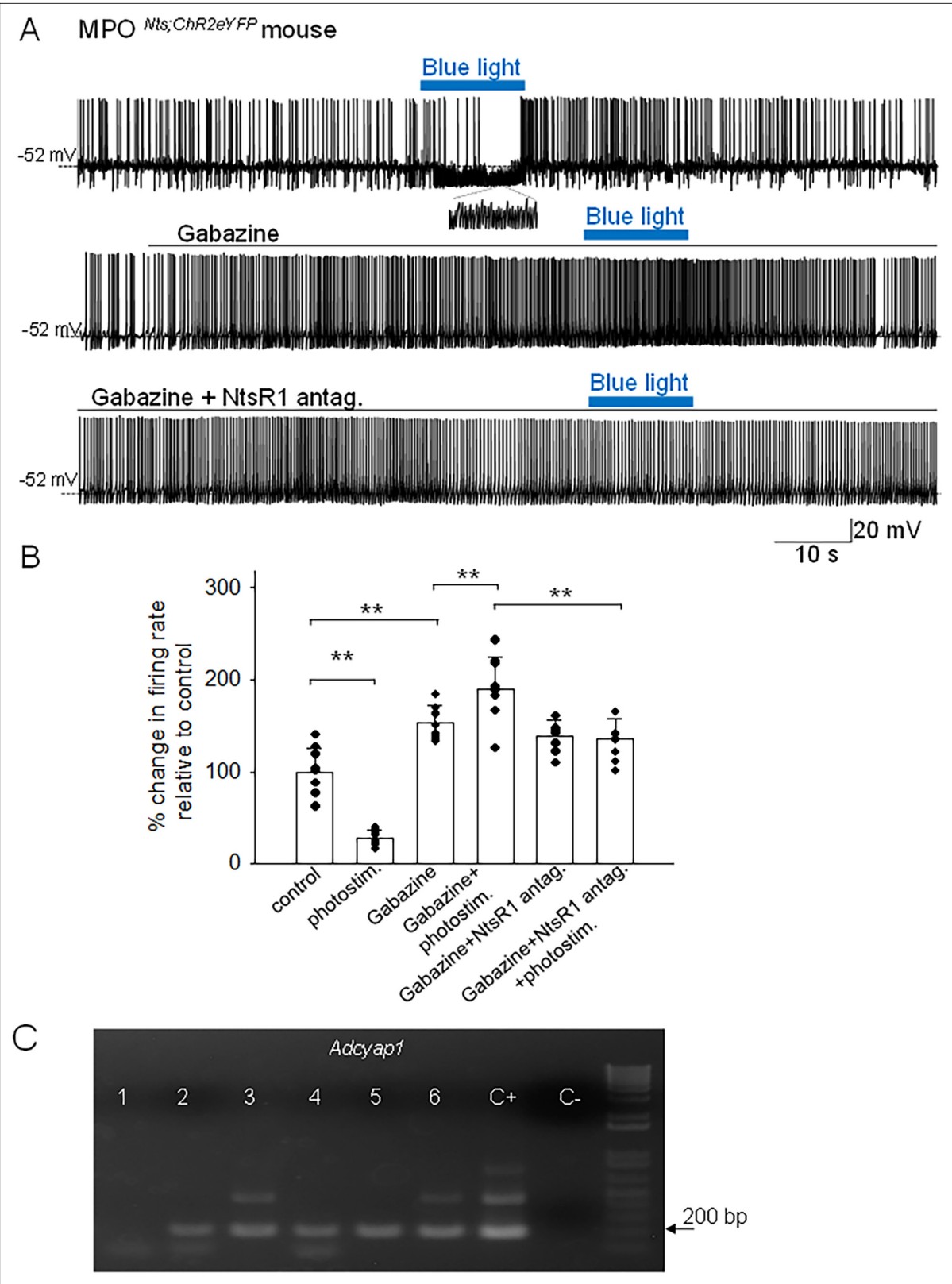

**Figure 2.** Effects of optogenetic stimulation of MPO$^{Nts;ChR2}$ neurons on the firing activity of nearby MPO neurons. (**A**) Optogenetic stimulation of a MPO$^{Nts;ChR2}$ neuron decreases the spontaneous firing rate of a nearby MPO neuron (upper trace) from 2.2 Hz to 0.9 Hz and increases the frequency of IPSC from 0.5 Hz to 21.8 Hz (see expanded trace). Gabazine (5 µM) (middle trace) increased the spontaneous firing activity of the neuron and abolished the activation of IPSCs by optogenetic stimulation. In the presence of Gabazine optogenetic stimulation increased the firing activity of the neuron from

*Figure 2 continued on next page*

*Figure 2 continued*

4.1 Hz to 9.6 Hz. This increase in firing activity was blocked by pe-incubation with the NtsR1 antagonist SR48692 (100 nM) (lower trace). (**B**) Bar charts summarizing the effects of optogenetic stimulation of MPO$^{Nts;ChR2}$ neurons on the spontaneous firing rates of nearby MPO neurons in control and in the presence of Gabazine (5 µM) and/or the NtsR1 antagonist SR48692 (100 nM). Bars represent means ± S.D. of the normalized firing rate relative to the control. The control value for the firing rate was calculated as the average value during the 5 min period preceding the optogenetic stimulation. Data pooled from n=9 neurons in each condition. There was a statistically significant difference between groups as determined by one-way ANOVA ($F_{(5,40)} = 77.71$, $p=1.08 \times 10^{-8}$) followed by Tukey's test between conditions; ** indicates statistical significance of $p<0.01$. The p-values of the Tukey's statistical comparisons among groups are presented in *Supplementary file 1-table 2*. (**C**) *Adcyap1* transcripts are present in MPO neurons inhibited by optogenetic stimulation of nearby MPO$^{Nts;ChR2}$ neurons. Representative results from six recorded MPO neurons. The expected size of the PCR product is 181 base pairs. Negative (−) control was amplified from a harvested cell without reverse-transcription, and positive control (+) was amplified using 1 ng of hypothalamic mRNA. *Adcyap1* transcripts were detected in five out of six neurons.

The online version of this article includes the following source data for figure 2:

**Source data 1.** Values of the percentage change in firing rates in each recorded cell during the treatments presented in *Figure 2B*.

**Source data 2.** Uncropped image of the gel presented in *Figure 2*.

induced a robust increase in the IPSCs frequency (*Figure 3A*). These results indicate that GABA release was not dependent neither upon excitatory synaptic transmission nor upon activation of NtsR1, that is it is synaptically released by MPO$^{Nts;ChR2}$ neurons. We have also found that brief photostimulation (10 s or less) resulted in increased frequency of IPSCs only (*Figure 3B and C*) while longer photostimulation (30 s or more) additionally activated an inward current (*Figure 3B*). The inward current was isolated in the presence of extracellular gabazine (5 µM) and averaged 9.26±2.39 pA (n=10). The inward current was abolished by bath application of NtsR1 antagonist SR48692 (100 nM) (*Figure 3B and D*). We have carried out a set of experiments in MPO$^{Nts;ChR2}$ from female mice and recorded similar results. Optogenetic stimulation resulted in a robust increase in the frequency of IPSCs followed, with a delay of 20–40 s, by an inward current that averaged 11.3±3.4 pA (n=5) (*Figure 3—figure supplement 1*). All the optogenetically evoked responses were blocked by incubation with gabazine (5 µM) and SR48692 (100 nM) (*Figure 3—figure supplement 1*).

Finally, we have recorded from MPO$^{Nts;ChR2}$ neurons and stimulated, using spot illumination, a different MPO$^{Nts;ChR2}$ neuron to question the presence of reciprocal connections. None out of 8 MPO$^{Nts;ChR2}$ neurons studied presented either IPSCs or an inward current evoked by optogenetic stimulation of other MPO$^{Nts;ChR2}$ neurons.

Thermoregulatory preoptic neurons project to the dorsomedial hypothalamus, arcuate, paraventricular thalamus and to the rostral raphe pallidus to control thermoeffector mechanisms (*Nakamura et al., 2022b*; *Tan et al., 2016*; *Hrvatin et al., 2020*). We have examined the brains of MPO$^{Nts;eYFP}$ mice in coronal slices to identify MPO projections to these sites, however no fluorescent signal was detected suggesting that MPO$^{Nts;eYFP}$ modulate CBT by acting on MPO$^{Adcyap1}$ neurons.

## Optogenetic stimulation of MPO$^{Nts;ChR2}$ neurons in vivo induces hyperthermia. In contrast, optogenetic stimulation of MPO$^{Nts;Slc32a1-/-;ChR2}$ neurons induces hypothermia

Optogenetic activation of *Nts* neurons in the MPO using MPO$^{Nts;ChR2}$ mice resulted in hyperthermia of up to 1.22 ± 0.35 °C when compared with control MPO$^{Nts;eYFP}$ mice (n=6 mice in each condition) (*Figure 4A and B*). To assess the role played by GABA release from MPO *Nts* neurons in the observed hyperthermia we activated *Nts* neurons in MPO$^{Nts;Slc32a1-;ChR2}$ mice (see Methods). Surprisingly, a decrease in CBT of 1.44 ± 0.29 °C was recorded relative to control MPO$^{Nts;Slc32a1-/-;eYFP}$ mice (*Figure 4C*).

Similar results were obtained also in female MPO$^{Nts;ChR2}$ mice and MPO$^{Nts;Slc32a1-/-;ChR2}$ mice. Intra-MPO optogenetic stimulation resulted in a hyperthermia of 1.40 ± 0.62 °C in MPO$^{Nts;ChR2}$ females and in a hypothermia of 1.63 ± 0.22 °C in MPO$^{Nts;Slc32a1-/-;ChR2}$ females (*Figure 4—figure supplement 1*).

## Optogenetic stimulation of MPO$^{Nts; Slc32a1-/-;ChR2}$ neurons increases the firing rate of MPO$^{Adcyap1}$ neurons by activating an inward current

The basal electrophysiological properties of MPO$^{Nts; Slc32a1-/-;ChR2}$ neurons were similar with those of MPO$^{Nts;ChR2}$ neurons. The neurons were spontaneously active, with an average firing rate at 36 °C of 4.79±2.45 Hz (n=26). Recordings from non-labeled MPO neurons revealed that spot illumination of

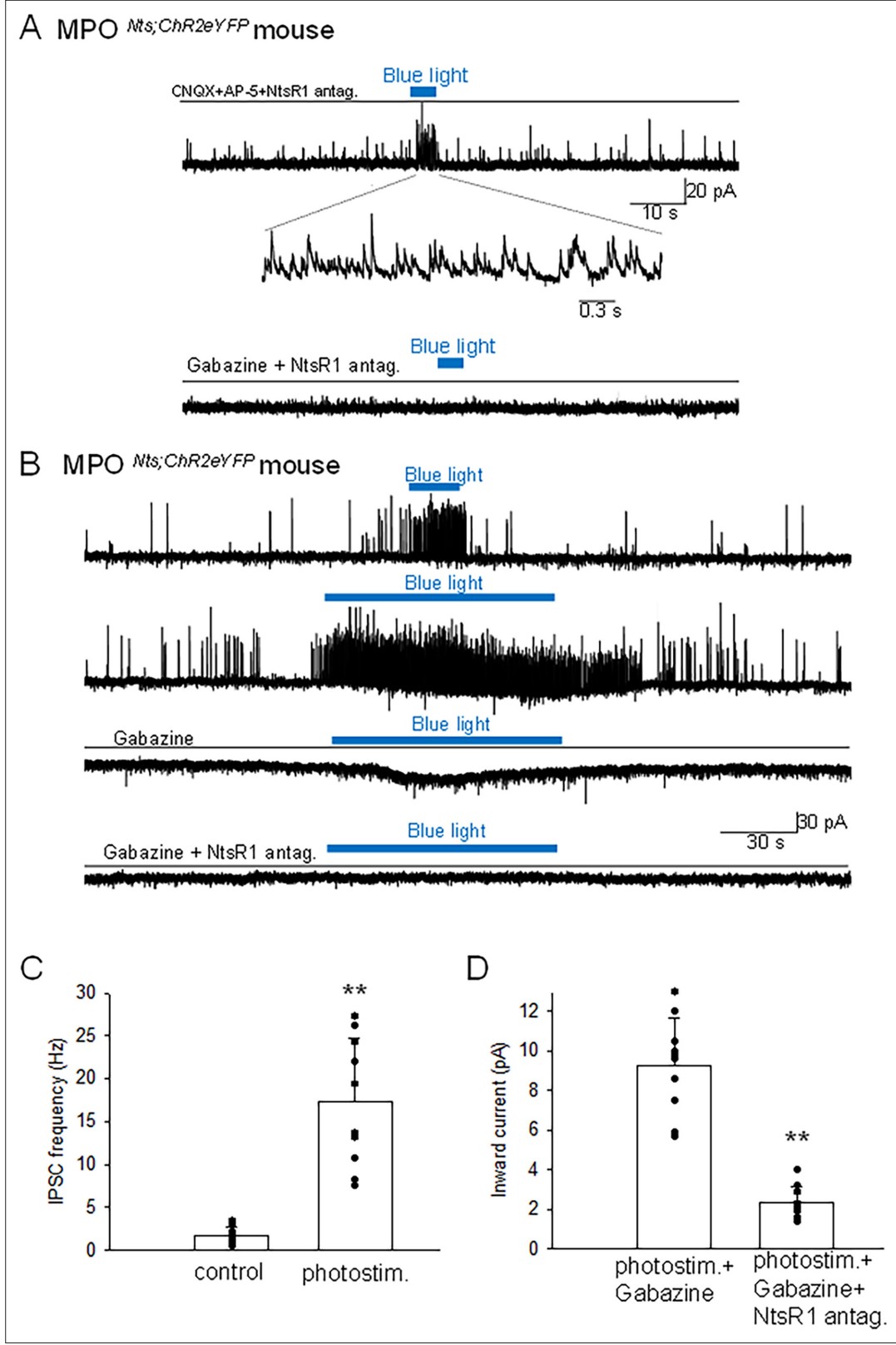

**Figure 3.** Optogenetic stimulation of MPO$^{Nts;ChR2}$ neurons increases the frequency of IPSCs and activates an inward current in nearby MPO neurons. (**A**) Optogenetic stimulation of a MPO$^{Nts;ChR2}$ neuron activates IPSCs in a nearby MPO neuron. Recordings were performed in the presence of CNQX (20 µM), AP-5 (50 µM) and the NtsR1 antagonist SR48692 (100 nM). The sIPSCs were abolished by Gabazine (5 µM) (lower trace). The neuron was

*Figure 3 continued on next page*

*Figure 3 continued*

held at –50 mV. (**B**) Optogenetic stimulation of the whole field of view containing several MPO^Nts;ChR2 neurons for 20 s activates IPSCs in a nearby MPO neuron (upper trace). Longer optogenetic stimulation (80 s) of the same neurons activated both IPSCs and an inward current (middle traces). The inward current was abolished by the NtsR1 antagonist SR48692 (100 nM) (lower trace). The neuron was held at –50 mV. (**C, D**) Bar charts summarizing the increase in the frequency of IPSCs (**C**) and the amplitude of the inward current (**D**) recorded in MPO neurons in response to optogenetic stimulation of several MPO^Nts;ChR2 neurons. (**C**) The IPSCs frequency increased from 1.75±0.97 Hz to 17.8±7.47 Hz in response to photostimulation (one-way ANOVA $F_{(1,18)}$=42.5, p=4 × 10^{-4}). The control value for the firing rate was calculated as the average value during the 5 min period preceding the optogenetic stimulation. (**D**) The average inward current activated by optogenetic stimulation decreased from 9.26±2.39 pA to 2.31±0.83 pA in the presence of the NtsR1 antagonist SR48692 (100 nM) (one-way ANOVA $F_{(1,18)}$=75.35, p=7.5 × 10^{-8}). Bars represent means ± S.D. Data pooled from n=10 neurons in each condition.

The online version of this article includes the following source data and figure supplement(s) for figure 3:

**Source data 1.** Change in IPSCs frequency and amplitude of inward current in response to photostimulation.

**Figure supplement 1—source data 1.** Change in IPSCs frequency and amplitude of inward current in response to photostimulation.

**Figure supplement 1.** Optogenetic stimulation of MPO^Nts;ChR2 neurons from females increases the frequency of IPSCs and activates an inward current in nearby MPO neurons.

nearby MPO^Nts;Slc32a1-/-;ChR2 neurons increased their firing rate (12 out of 26 neurons studied) or had no effect in the others (14 out of 26 neurons). *Figure 5A* illustrates the increase in firing rate of a MPO neuron (from 1.6 Hz to 3.9 Hz) induced by optogenetic stimulation of a nearby MPO^Nts;ChR2 neuron, an effect which was associated with an apparent depolarization. Overall, optogenetic stimulation of nearby MPO^Nts;Slc32a1-/-;ChR2 neurons increased the firing rate of the recorded MPO neurons from 5.68±1.66 Hz to 12.20±3.65 Hz (n=12) (*Figure 5B*). This effect was blocked by preincubation with the NtsR1 antagonist SR48692 (100 nM; *Figure 5A and B*). In voltage-clamp experiments photostimulation activated an inward current which was abolished by bath application of NtsR1 antagonist SR48692 (100 nM) (*Figure 5C and D*). Using s.c. RT/PCR assays we have detected *Adcyap1* transcripts in 8 out of 11 neurons excited by photostimulation tested.

We have studied the presence of *Slc32a1* and *Nts* transcripts in MPO slices from *Nts^cre Slc32a1^lox/lox* mice and in control *Nts^cre Slc32a1^+/+* mice (*Figure 5—figure supplement 1*). *Nts* and *Slc32a1* transcripts overlapped in control tissue while there was no colocalization in tissue from *Nts^cre Slc32a1^lox/lox* mice (*Figure 5—figure supplement 1*).

## Characterization of the fever response, the heat exposure and the cold exposure responses as well as of the circadian CBT profile in *Nts^Slc32a1-/-* mice

Since MPO^Nts;Slc32a1-/-;ChR2 mice displayed drastically different CBT responses to optogenetic stimulation relative to those of MPO^Nts;ChR2 mice, suggesting that GABA release by *Nts* neurons played an important role in thermoregulation, we decided to further characterize thermoregulatory responses in the *Nts^Slc32a1-/-* mice.

The LPS-induced fever was significantly different in *Nts^Slc32a1-/-* mice when compared with that of wild-type littermates *Nts^cre Slc32a1^+/+* (*Figure 6A*). During the intermediate and late phases of fever, the CBT was significantly lower in *Nts^cre Slc32a1^lox/lox* mice (*Figure 6A*).

During heat exposure the CBT increased at a faster rate in *Nts^cre Slc32a1^lox/lox* mice than in *Nts^cre Slc32a1^+/+* mice (*Figure 6B*). The *Nts^cre Slc32a1^lox/lox* mice and *Nts^cre Slc32a1^+/+* mice reached 41.5 °C within 1.33±0.04 hr and 1.53±0.04 hr, respectively (Kruskal-Wallis, p=2.14 × 10^{-4}, n=9 mice in each group). The *Nts^cre Slc32a1^lox/lox* mice displayed significantly higher CBT (by ~0.5 °C) after 45 min of heat exposure (*Figure 6B*).

The circadian CBT profile of *Nts^cre Slc32a1^lox/lox* mice was also altered relative to that of *Nts^cre Slc32a1^+/+* mice: the active phase started ~1 hr earlier and ended ~3 hr later (*Figure 6C*).

During a 3 hr exposure in a cold room at 4°C the two mouse lines displayed strikingly different CBT responses (*Figure 6D*). The control mice displayed a gradual decrease in CBT which stabilized at 33-34 °C by third hour of the cold exposure. In contrast, *Nts^cre Slc32a1^lox/lox* mice displayed a hyperthermia of 37.4°C at the beginning of the experiment followed by a slow decline that reached ~35 °C

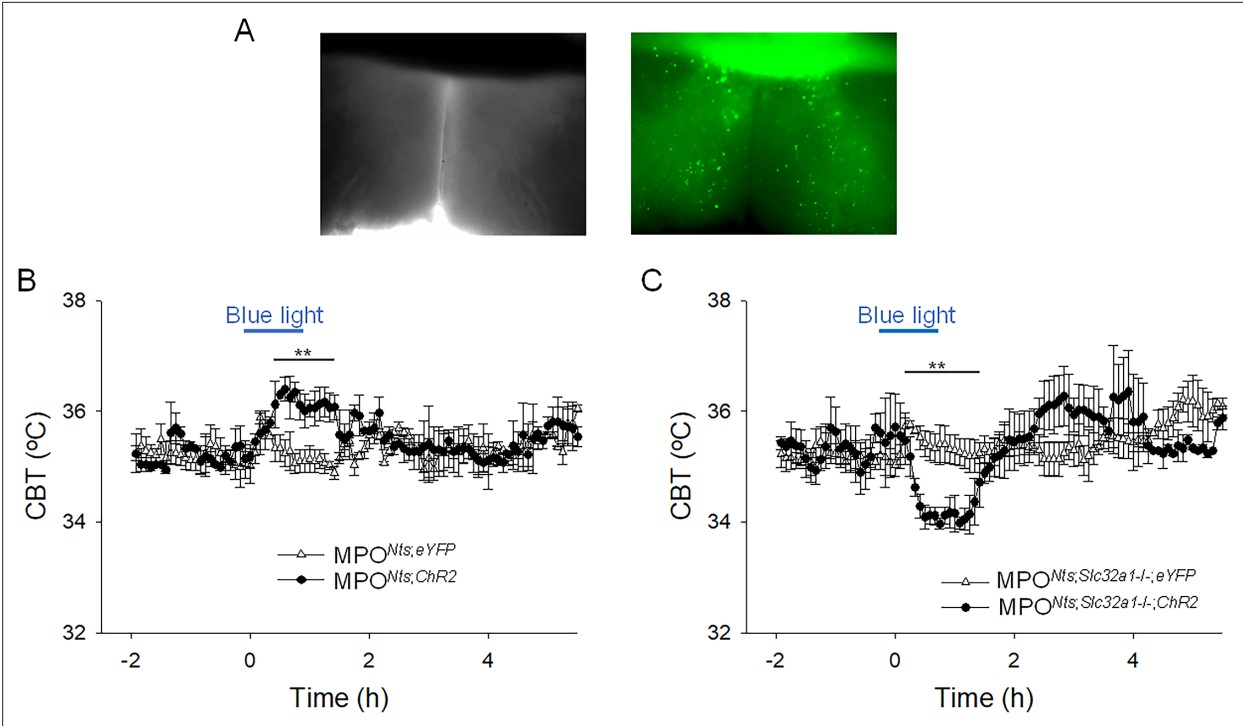

**Figure 4.** Optogenetic activation of MPO$^{Nts;ChR2}$ neurons induces hyperthermia while optogenetic activation of MPO$^{Nts;Slc32a1-/-;ChR2}$ neurons induces hypothermia. (**A**) DIC (left) and fluorescence (right) images of an acute slice from MPO$^{Nts;ChR2}$ mouse indicating hChR2-eYFP expression in the MPO. (**B**) Optogenetic stimulation of MPO$^{Nts;ChR2}$ neurons (●) in vivo for 1 hr (blue light) induced a hyperthermia of 1.22 ± 0.35 °C relative to control (Δ). The response was statistically different to the response to photostimulation of control MPO$^{Nts;eYFP}$ mice (Δ) (one-way repeated measures ANOVA, $F_{(1,111)}=20.9$, $p=1.2 \times 10^{-5}$, followed by Mann-Whitney U tests for each time point, ** $p<0.01$). (**C**) Optogenetic stimulation of MPO$^{Nts;Slc32a1-/-;ChR2}$ neurons (●) in vivo for 1 hour (blue light) induced a hypothermia of 1.44 ± 0.29 °C relative to control (Δ). The response was statistically different to the response to photostimulation of control MPO$^{Nts;Slc32a1-/-;eYFP}$ mice (Δ) (one-way repeated measures ANOVA, $F_{(1,112)}=8.27$, $p=4.8 \times 10^{-3}$, followed by Mann-Whitney U tests for each time point, ** $p<0.01$). (**B,C**) The points represent averages ± S.D. through the 7 hr recording period. Experiments were carried out in parallel in groups of six mice.

The online version of this article includes the following source data and figure supplement(s) for figure 4:

**Source data 1.** Change in CBT in response to photostimulation.

**Figure supplement 1.** In female mice optogenetic activation of MPO$^{Nts;ChR2}$ neurons induces hyperthermia while optogenetic activation of MPO$^{Nts;Slc32a1-/-;ChR2}$ neurons induces hypothermia.

**Figure supplement 1—source data 1.** Change in CBT in response to photostimulation.

at the end of the three hours. Upon return to vivarium the CBT of controls returned to 37 °C while the *Nts$^{cre}$ Slc32a1$^{lox/lox}$* mice displayed a transient hyperthermia to 39°C.

## Activation of central *Nts$^{cre}$ hM3D$^{lox/lox}$* neurons induces potent hypothermia

Previous studies have reported the intracerebroventricular or intra MPO infusions of exogenous Nts induce potent hypothermia (*St-Gelais et al., 2006*; *Tabarean, 2020*). Since our results indicated that activation of MPO Nts neuron does not mimic this effect we hypothesized that Nts release by Nts neurons in other brain regions may be required to induce hypothermia. In order to activate all central Nts neurons we have generated *Nts$^{cre}$ hM3D$^{lox/lox}$* mice (see Materials and methods) and employed chemogenetic stimulation. Activation of central *Nts$^{cre}$ hM3D$^{lox/lox}$* neurons induced a potent hypothermia of 4.8 ± 0.6 °C relative to *Nts-cre* mice (control) (*Figure 7A*), value comparable with that obtained with MPO infusions of exogenous Nts (*Tabarean, 2020*). To test the hypothesis that the main locus of action is the MPO and to determine the possible role of NtsR1 and NtsR2 in this effect we have performed the chemogenetic activation of *Nts$^{cre}$ hM3D$^{lox/lox}$* in the presence of NtsR1 and/ or NtsR2 antagonists in the MPO (*Figure 7B*). The antagonists (or aCSF as control) were infused via

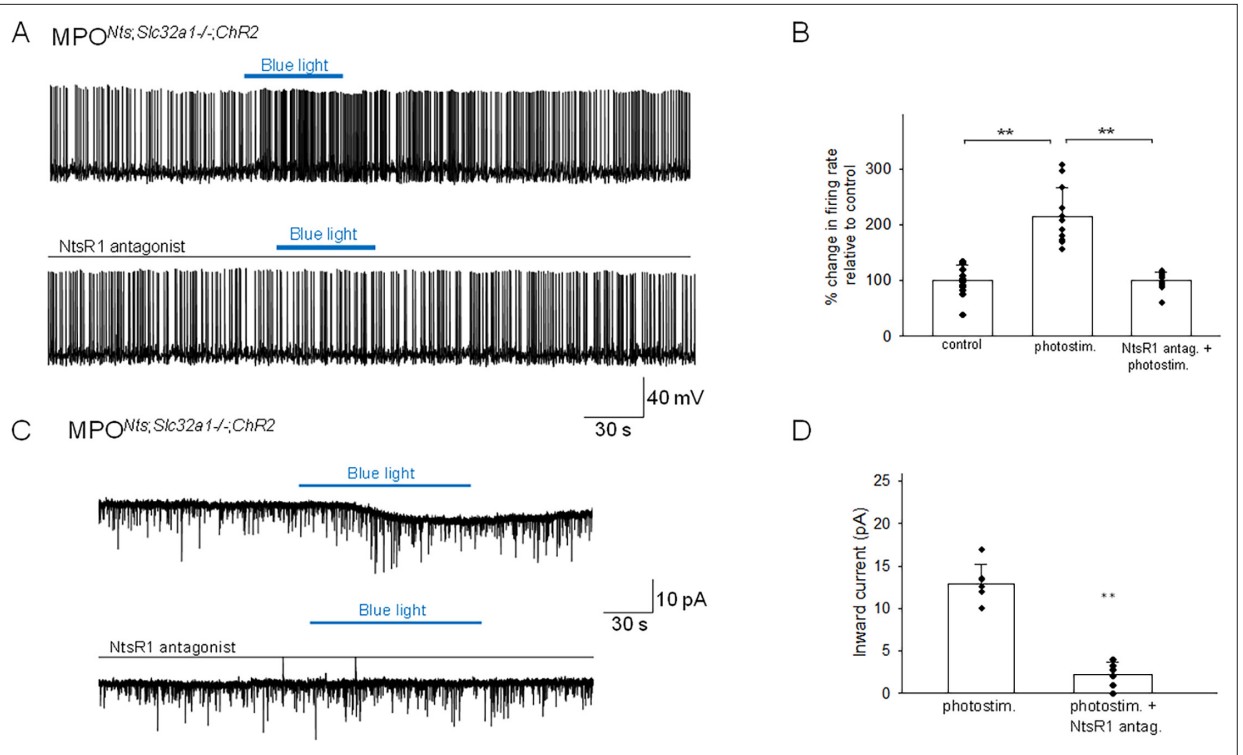

**Figure 5.** Optogenetic stimulation of MPO$^{Nts;Slc32a1-/-;ChR2}$ neurons increases the firing activity of nearby MPO neurons. (**A**) Optogenetic stimulation of MPO$^{Nts;Slc32a1-/-;ChR2}$ neurons increases the spontaneous firing rate of a nearby MPO neuron (upper trace) from 1.6 Hz to 3.9 Hz. The photostimulation-induced increase in firing activity was blocked by pe-incubation with the NtsR1 antagonist SR48692 (100 nM) (lower trace). (**B**) Bar charts summarizing the effects of optogenetic stimulation of MPO$^{Nts;Slc32a1-/-;ChR2}$ neurons on the spontaneous firing rates of nearby MPO neurons in control and in the presence of the NtsR1 antagonist SR48692 (100 nM). Bars represent means ± S.D. of the normalized firing rate relative to the control. Data pooled from n=12 neurons in each condition. There was a statistically significant difference between groups as determined by one-way (ANOVA F(2,22) = 42.45, p=2.80 × 10$^{-8}$) followed by Tukey's test between conditions; ** indicates statistical significance of p<0.01, * indicates p<0.05. The p-values of the Tukey's statistical comparisons among groups are presented in ***Supplementary file 1-table 3***. (**C**) Optogenetic stimulation of the whole field of view containing several MPO$^{Nts;ChR2}$ neurons for 20 s activates IPSCs in a nearby MPO neuron (upper trace). Longer optogenetic stimulation (80 s) of the same neurons activated both IPSCs and an inward current (middle traces). The inward current was abolished by the NtsR1 antagonist SR48692 (100 nM) (lower trace). The neuron was held at –50 mV. (**D**) Bar chart summarizing the amplitude of the inward current recorded in MPO neurons in response to optogenetic stimulation in control and during incubation with the NtsR1 antagonist SR48692 (100 nM). The average inward current activated by optogenetic stimulation decreased from 12.86±2.33 pA to 2.18±1.49 pA in the presence of MPO$^{Nts;Slc32a1-/-;ChR2}$ neurons the NtsR1 antagonist SR48692 (100 nM) (one-way ANOVA F(1,22)=193.73, p=2.49 × 10$^{-8}$). Bars represent means ± S.D. Data pooled from n=12 neurons.

The online version of this article includes the following source data and figure supplement(s) for figure 5:

**Source data 1.** Change in firing rate and amplitude of inward current in response to photostimulation.

**Figure supplement 1.** Expression of *Slc32a1* transcripts in preoptic slices from *Nts$^{cre}$ Slc32a1$^{lox/lox}$* and *Nts$^{cre}$ Slc32a1$^{+/+}$* male mice.

**Figure supplement 1—source data 1.** Average number of Slc32a1 positive cells in *Nts$^{cre}$ Slc32a1$^{+/+}$* and *Nts$^{cre}$ Slc32a1$^{lox/lox}$* tissue.

a bilateral guide cannula 1.5 hr prior to CNO injection (i.p. 1 mg/kg). The NtsR1 antagonist SR48692 (300 nM, 100 nl, blue trace) decreased the chemogenetically induced hypothermia by 41% relative to control (***Figure 7B***). Higher concentrations of SR48692 (600 nM and 1.5 µM) resulted in similar reductions in the hypothermic response of 43 and 31%, respectively (n=6 mice in each condition). Following intra-MPO injections of NTSR2 antagonist NTRC 824 (200 nM, 100 nl, black trace) the CNO-induced hypothermia was decreased by 32% relative to control (***Figure 7B***). When the two doses of antagonists were infused together the chemogenetically-induced hypothermia decreased by 76% (***Figure 7B***, red trace). These results indicate that the MPO accounts for most of the hypothermia induced by *Nts$^{cre}$ hM3D$^{lox/lox}$* neurons and that both receptor subtypes play an important role.

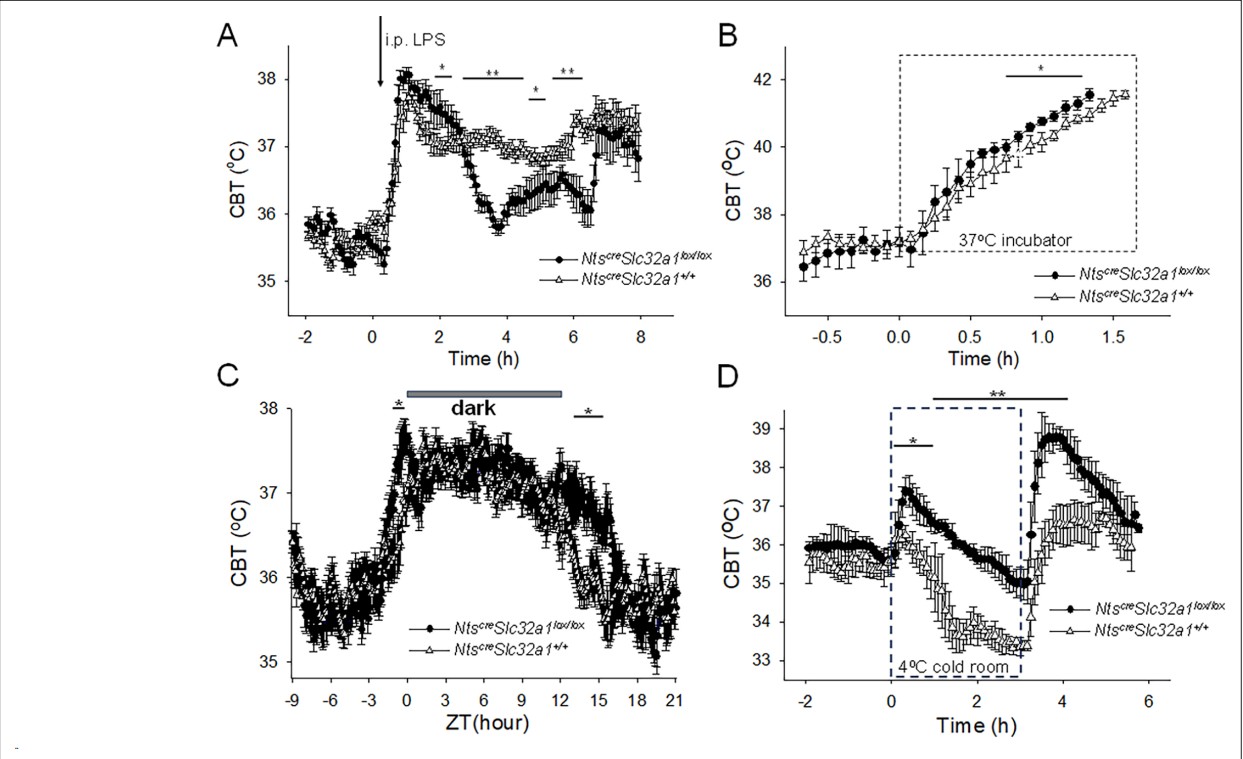

**Figure 6.** Altered thermoregulatory profile of *Nts^cre Slc32a1^lox/lox* mice. (**A**) CBT responses to i.p. injection (arrow) of LPS (0.03 mg/kg) in *Nts^cre Slc32a1^+/+* mice (Δ, control) and *Nts^cre Slc32a1^lox/lox* mice (●). LPS induced fever responses with differential profiles (one-way repeated measures ANOVA, $F_{(1,94)}=9.31$, $p=2.948 \times 10^{-3}$, followed by Mann-Whitney U tests for each time point, ** $p<0.01$, * $p<0.05$). (**B**) CBT responses during a heat test in an incubator at 37°C. The CBT increased faster in *Nts^cre Slc32a1^lox/lox* mice (●) than in *Nts^cre Slc32a1^+/+* mice (Δ, control) (repeated measures ANOVA, $F_{(1,16)} = 33.49$, $p=2.78 \times 10^{-5}$, followed by Mann-Whitney U tests for each time point, * $p<0.05$). (**C**) Circadian CBT profiles in *Nts^Slc32a1-/-* mice (●) than in *Nts^cre Slc32a1^+/+* mice (Δ, control). *Nts^cre Slc32a1^lox/lox* mice display a longer active phase relative to controls (repeated measures ANOVA, $F_{(1,359)} = 173.35$, $p=1.10 \times 10^{-9}$, followed by Mann-Whitney U tests for each time point, * $p<0.05$). Data for each mouse represents the average of 10 different 24 hr periods. (**D**) CBT responses during a cold test in an incubator at 4 °C. *Nts^cre Slc32a1^lox/lox* mice (●), in contrast with *Nts^cre Slc32a1^+/+* mice (Δ, control), displayed a significant hyperthermia at the beginning as well as following the end of the cold exposure (repeated measures ANOVA, $F_{(1,92)}=220.89$, $p=4.3 \times 10^{-14}$, followed by Mann-Whitney U tests for each time point, * $p<0.05$, ** $p<0.01$). (**A–D**) The points represent averages ± S.D. (n=6 male mice).

The online version of this article includes the following source data for figure 6:

**Source data 1.** Differential CBT responses to LPS, heat and cold exposure in *Nts^cre Slc32a1^+/+* and *Nts^cre Slc32a1^lox/lox* mice.

## Chemogenetic activation of *Nts^cre hM3D^lox/lox* neurons and projections in the MPO results in potent excitation of *MPO^Adcyap1*

To understand the mechanisms by which chemogenetic activation of *Nts^cre hM3D^lox/lox* neurons induced hypothermia, we have studied the effect of CNO on the activity of MPO non-labeled neurons in acute slices from *Nts^cre hM3D^lox/lox* mice. Bath application of CNO excited 9 out 20 neurons studied and had no effect in the others. *Figure 7C* illustrates such a response in a non-labeled MPO neuron. Interestingly prior to depolarization an increase in the frequency of sIPSPs (arrow) was recorded. The average firing rate increased from 4.15±2.95 Hz to 10.63±5.83 Hz (n=9, ANOVA, $F_{(1/16)}=8.99$ $p=8.52 \times 10^{-3}$) in response to CNO (3 μM) (*Figure 7D*). S.c. RT/PCR assays identified *Adcyap1* transcripts in seven of the nine neurons excited by CNO tested. In voltage-clamp experiments bath application of CNO (3 μM) activated an inward current and potently increased the frequencies and amplitudes of both sEPSCs and sIPSCs (*Figure 7E*). The frequencies of sEPSCs and sIPSCs increased from 2.76±1.16 Hz to 9.14±5.36 Hz and 1.98±1.43 Hz to 8.84±6.38 Hz, respectively (n=6) (*Figure 7F and H*). The amplitudes of sIPSCs and sEPSCs increased from 13.61±4.51 pA to 29.60±5.32 pA and 13.02±3.99 pA to 22.12±5.53 pA, respectively (n=6)(*Figure 7G,I*). The NtsR1 antagonist SR48692 (100 nM) abolished the inward current activated by CNO (*Figure 7E*, middle trace) and significantly reduced the effect on

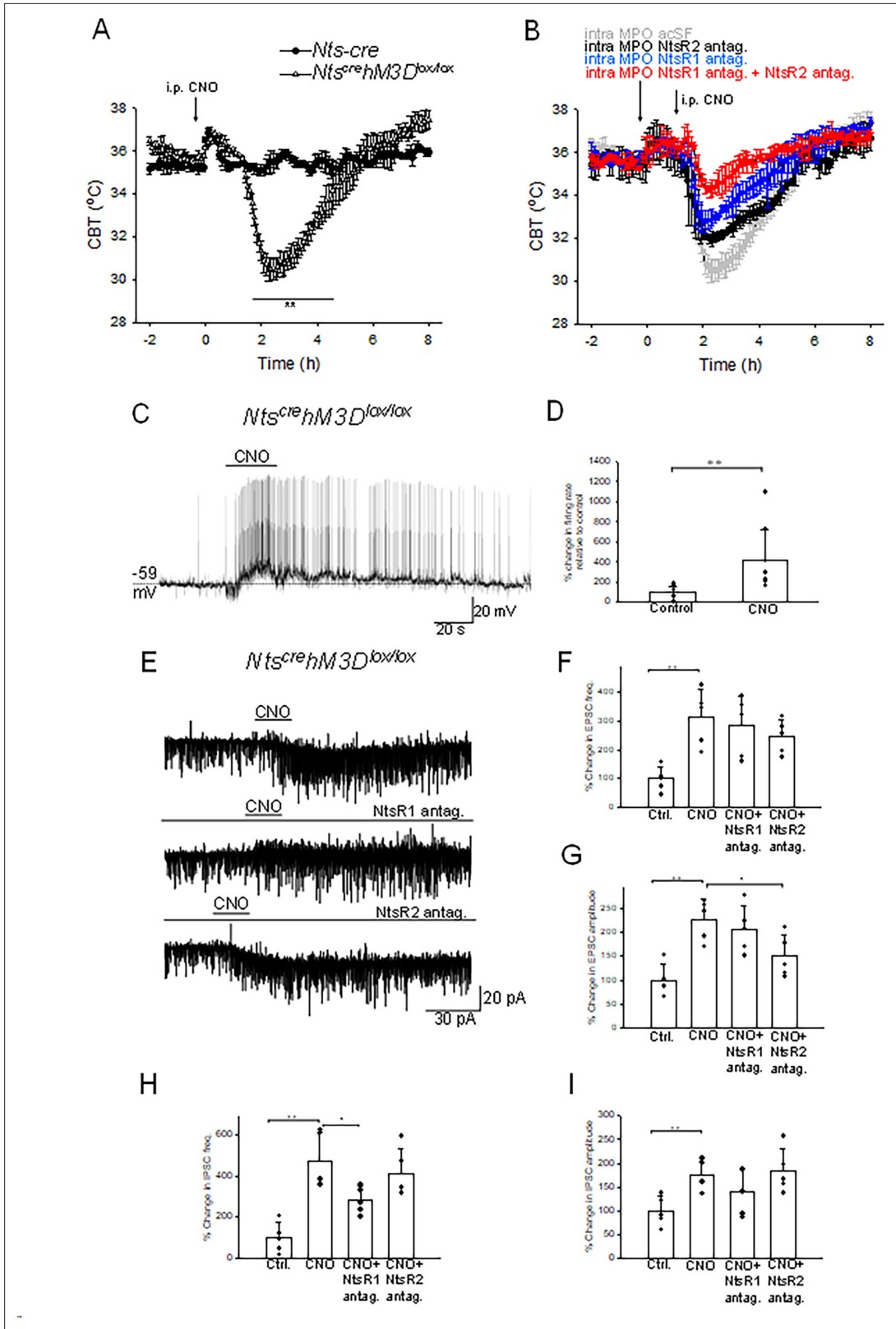

**Figure 7.** Chemogenetic activation of neurotensinergic neurons and projections induces hypothermia and potently excites MPO neurons. (**A**) I.p. injection (arrow) of CNO (20 mM, 3 µl) in $Nts^{cre} hM3D^{lox/lox}$ mice (Δ) and in $Nts$-$cre$ mice (control, ●). CNO induced a hypothermia of 4.8 ± 0.6 °C (repeated measures ANOVA, $F(1,242)=21.72$, $p=2.8 × 10^{-6}$, followed by followed by Mann-Whitney U tests for each time point, ** $p<0.01$). (**B**) Role of NtsR1 and NtsR2 expressed in the MPO in the CNO-induced activation of $Nts^{cre}hM3D^{lox/lox}$ neurons. $Nts^{cre}hM3D^{lox/lox}$ mice received a bilateral

*Figure 7 continued on next page*

*Figure 7 continued*

infusion of aCSF (gray), NtsR1 antagonist SR48692 (300 nM, 100 nl, blue), NtsR2 antagonist NTRC 824 (200 nM, 100 nl, black) and NtsR1 antagonist (300 nM, 100 nl)+NtsR2 antagonist (200 nM, 100 nl) (red) 1.5 hr prior to an i.p. injection of CNO (20 mM, 3 µl). The antagonists significantly reduced the hypothermia (repeated measures ANOVA, $F_{(3,360)}=71.33$, $p=8.82 \times 10^{-16}$). (**A,B**) The points represent averages ± S.D. Experiments were carried out in parallel in groups of 6. (**C**) Chemogenetic activation of neurotensinergic neurons and neurotensinergic projections in the MPO by bath application of CNO (3 µM) in slices from $Nts^{cre}hM3D^{lox/lox}$ mice depolarizes and increases the firing rate of a MPO neuron. The firing rate increased from 0.15 Hz to 1.65 Hz. (**D**) Bar chart summarizing the effect of chemogenetic activation of neurotensinergic neurons and projections in the MPO on the firing rates of MPO neurons. There was a statistically significant difference between groups as determined by one-way ANOVA ($F_{(1,16)}=8.99$, $p=8.52 \times 10^{-3}$; ** indicates statistical significance of $p<0.01$). Bars represent means ± S.D. of the normalized firing rate relative to the control. Data pooled from n=10 neurons. (**E**) Chemogenetic activation of neurotensinergic neurons and neurotensinergic projections in the MPO by bath application of CNO (3 µM) in slices from $Nts^{cre}hM3D^{lox/lox}$ mice increases the amplitudes and frequencies of both IPSCs and EPSCs and activates an inward current (upper trace). The NtsR1 antagonist SR48692 (100 nM) abolished the inward current activated by CNO (middle trace). The NtsR2 antagonist NTRC 824 (100 nM) did not change the inward current activated by CNO but significantly decreased the amplitude of sEPSCs (lower trace). The neuron was held at –50 mV. (**F, G, H, I**) Bar charts summarizing the increase in sEPSCs frequency (**F**) and amplitude (**G**) and IPSCs frequency (**H**) and amplitude (**I**) Bars represent means ± S.D. of the normalized frequency relative to the control. Data pooled from n=6 neurons. The changes were statistically significant for sEPSCs frequency (one-way ANOVA $F_{(3,19)}=6.94$, $p=3.33 \times 10^{-3}$) and amplitude (one-way ANOVA $F_{(3,19)}=9.20$, $p=9.03 \times 10^{-4}$) as well as for sIPSCs frequency (one-way ANOVA $F_{(3,19)}=12.61$, $p=1.71 \times 10^{-4}$) and amplitude (one-way ANOVA $F_{(3,19)}=4.53$, $p=1.76 \times 10^{-3}$). The P values for the inter-group comparisons are listed in *Supplementary file 1-tables 4-7*.

The online version of this article includes the following source data for figure 7:

**Source data 1.** Chemogenetic stimulation of $Nts^{cre}hM3D^{lox/lox}$ neurons induces changes in CBT, firing rates, and in the amplitude and frequency of synaptic currents.

the frequency of sIPSCs (*Figure 7H*). The NtsR2 antagonist NTRC 824 (100 nM) did not change the inward current activated by CNO but significantly decreased the amplitude of sEPSCs (*Figure 7G*).

## Discussion

In this study we have characterized a novel population of MPO neurons that express Nts (MPO$^{Nts}$). These neurons are GABAergic as indicated by detection of *Slc32a1* transcripts as well as by the activation of gabazine-sensitive IPSCs in postsynaptic neurons in response to spot illumination of MPO$^{Nts;ChR2}$ neurons. Interestingly, this population had a tendency to burst in response to depolarization, a characteristic of neurosecretory neurons (*Armstrong et al., 2010*; *Black et al., 2014*). In response to optogenetic stimulation MPO$^{Nts;ChR2}$ neurons released both GABA and Nts as indicated by pharmacological experiments. Postsynaptically, increased frequency of IPSCs was detected instantaneously upon photostimulation while an inward current was activated after 20–40 s, indicating a differential timecourse of action. In view of its slower timecourse of action, Nts may act to limit and/or shorten the inhibitory effect of GABA on the activity of the postsynaptic neurons following increased burst of activity of MPO$^{Nts}$ neurons. Nevertheless, the net effect during photostimulation was a decrease in the firing rate of the postsynaptic neuron, an increase in firing rate being unmasked only upon ending photostimulation or in the presence of gabazine. Recordings from MPO$^{Nts;Slc32a1-/-;ChR2}$ neurons confirmed the activation of an inward current in postsynaptic neurons in response to photostimulation. The inhibition-excitation sequence observed here is compatible with a scenario where the postsynaptic neuron is inhibited if the presynaptic neurotensinergic neurons are active and quickly activated upon the end of this presynaptic spiking. In more general terms, a temporal sequence where fast inhibition precedes a slower excitation allows to quickly reset the membrane potential of the postsynaptic neurons making them ready to respond to new stimuli. The initial inhibition can also reset or synchronize neural activity, allowing the subsequent excitation to produce more precisely timed spiking assuming the process occurs simultaneously across a population of neurons. Alternatively, the sequence of inhibition followed by excitation can contribute to generating and maintaining rhythmic activity in neural networks and create oscillatory patterns at specific frequencies, depending on those of the synaptic input. Finally, the overall responsiveness of a neural circuit can be influenced by rapid inhibition to reduce gain followed by excitation to increase it.

In terms of local neuronal networks, MPO$^{Nts}$ neurons did not present reciprocal connections instead, they appeared to synapse onto MPO$^{Adcyap1}$ neurons. *Adcyap1* has been identified as a marker of preoptic thermoregulatory neurons activated in a warm environment (*Tan et al., 2016*) and also during torpor (*Hrvatin et al., 2020*). Activation of MPO$^{Adcyap1}$ neurons induces hypothermia (*Tan et al.,*

2016; *Hrvatin et al., 2020*). However, *Adcyap1* is expressed in several subpopulations of neurons, representing a large proportion of the entire preoptic population (*Moffitt et al., 2018*; *Hrvatin et al., 2020*). Recent studies have identified a subpopulation of the MPO$^{Adcyap1}$ neurons that expresses the EP3 prostanoid receptors, is activated in warm environment and inhibited in a cold environment (*Nakamura et al., 2022b*). This population plays a critical role in the initiation of the fever response (*Machado et al., 2020*; *Nakamura et al., 2022b*). Modulating the activity of preoptic EP3 neurons up or down induces hypothermia or hyperthermia, respectively (*Nakamura et al., 2022b*). It is possible that MPO$^{Nts}$ neurons overlap with a previously hypothesized population of inhibitory neurons that receive peripheral cold input and project to EP3 neurons to trigger thermogenesis and inhibit heat loss (*Morrison and Nakamura, 2019*; *Nakamura et al., 2022a*).

In view of the potent hypothermic properties of Nts when applied peripherally, centrally or in the preoptic area it was surprising to find that photostimulation of MPO$^{Nts;ChR2}$ neurons in vivo resulted in a mild hyperthermia, instead. In contrast, photostimulation of MPO$^{Nts;Slc32a1-/-;ChR2}$ neurons, which caused a net excitation of MPO$^{Adcyap1}$ neurons, resulted in hypothermia, supporting the idea that their excitation induces hypothermia (*Tan et al., 2016*; *Hrvatin et al., 2020*).

Numerous studies have revealed that, in the preoptic area, GABA exerts complex effects on CBT depending on the environmental context and the specific neuronal subpopulation involved (*Frosini et al., 2004*; *Ishiwata et al., 2005*; *Osaka, 2004*; *Zhao et al., 2017*). *Nts$^{cre}$ Slc32a1$^{lox/lox}$* mice displayed significantly different thermoregulatory characteristics suggesting that GABA release from neurotensinergic neurons plays an important role in the respective responses. The fever response to endotoxin, when compared with the control, was characterized by lower CBT during the intermediate and late phases of the fever suggesting that GABA signaling is involved in the hyperthermic mechanisms activated. This finding is in line with the previous reports that preoptic GABA can induce thermogenesis (*Ishiwata et al., 2005*; *Osaka, 2004*); however, it is in disagreement with observations that the induction of fever is associated with decreased synaptic inhibition in the preoptic area (*Osaka, 2008*; *Tabarean et al., 2004*). It is possible that both mechanisms influence CBT during fever with the former playing a prevalent role in the intermediate and later phases.

In a warm environment preoptic release of GABA decreases and exogenous GABA-A antagonists applied locally have hyperthermic effects (*Ishiwata et al., 2005*), thus *Nts$^{cre}$ Slc32a1$^{lox/lox}$* would be expected to have a higher CBT during a heat challenge. Indeed, we observed that during heat exposure the CBT of *Nts$^{cre}$ Slc32a1$^{lox/lox}$* mice increased significantly faster than in the controls suggesting that in a warm environment GABA release from neurotensinergic neurons is involved in the control of heat loss mechanisms.

During cold exposure *Nts$^{cre}$ Slc32a1$^{lox/lox}$* mice displayed higher CBT relative to controls. Previous studies have revealed increased endogenous release of GABA in the preoptic area during cold exposure and hypothermic effects of exogenous GABA-A antagonist in a cold environment (*Ishiwata et al., 2005*). Thus, a decreased release of GABA from *Nts$^{cre}$ Slc32a1$^{lox/lox}$* neurons would be expected to induce a larger hypothermia during cold exposure, the opposite of the observed results. However, while an increase in preoptic GABA concentration induces hyperthermia, at all ambient temperatures, a widespread central increase in GABA is hypothermic (*Frosini et al., 2004*; *Nikolov and Yakimova, 2011*), therefore it is possible that *Slc32a1* deletion in Nts neurons outside of the MPO may impact CBT during cold exposure and/or that the respective regions are differentially recruited during distinct thermoregulatory mechanisms.

Finally, *Nts$^{cre}$ Slc32a1$^{lox/lox}$* mice have entered the 'up' phase of the circadian CBT rhythm earlier and ended it later when compared to the control, supporting a role of neurotensinergic neurons, possibly in the suprachiasmatic nucleus, in the control of circadian rhythm (*Meyer-Spasche et al., 2002*; *Yamada et al., 1995*).

To question whether stimulation of all central neurotensinergic neurons induces hypothermia, we have generated *Nts$^{cre}$ hM3D$^{lox/lox}$* mice. Chemogenetic activation of central *Nts$^{cre}$ hM3D$^{lox/lox}$* neurons resulted in robust hypothermia, comparable to that induced by central infusion of exogenous neurotensin. The response was largely blocked by NtsR1 and NtsR2 antagonists injected in the MPO, suggesting that the locus of action is the MPO. Taken together our results suggest that stimulation of neurotensinergic neurons in the MPO is not sufficient to induce hypothermia although this region represents the locus where the hypothermia is triggered. The amplitude and duration of the hypothermia oberved when activating all central *Nts$^{cre}$ hM3D$^{lox/lox}$* neurons is reminiscent of the

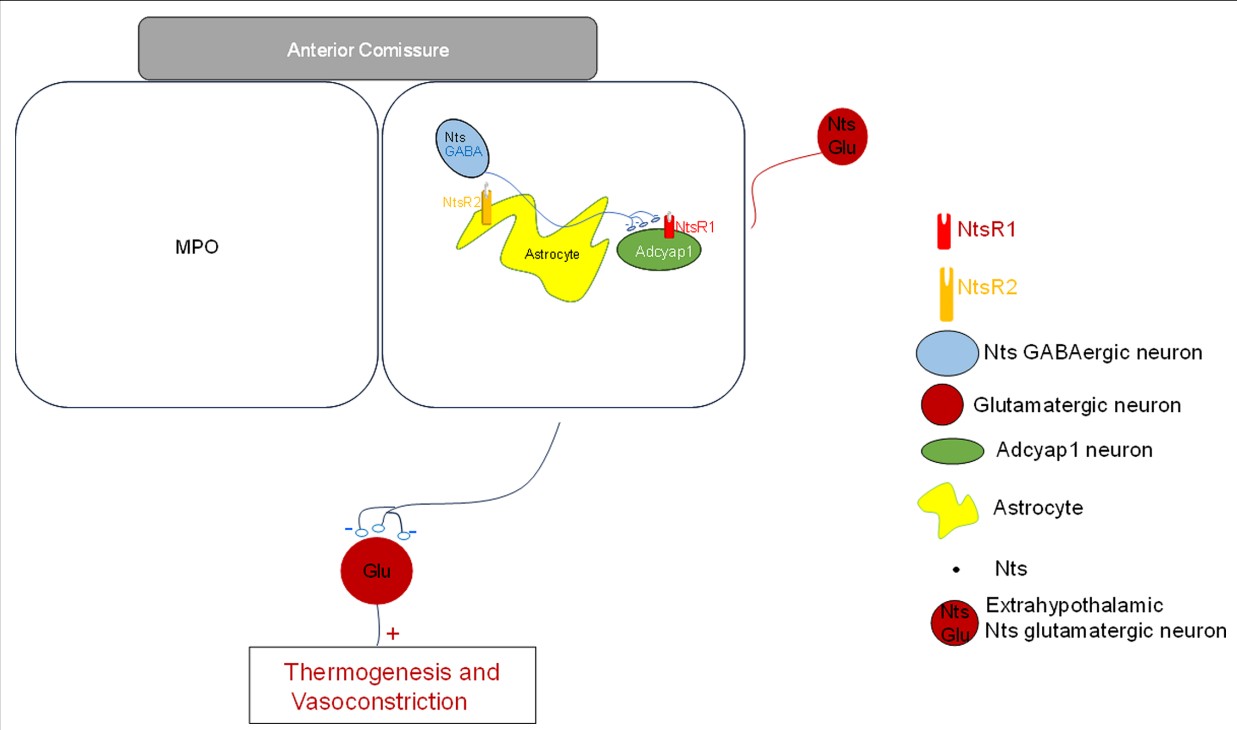

**Figure 8.** Schematic representation of neurotensinergic neurons in a thermoregulatory preoptic pathway. Preoptic thermoregulatory *Adcyap1* neurons, assumed to be glutamatergic, project to inhibitory interneurons in other brain regions that project to neurons controlling thermogenesis and vasoconstriction. *Adcyap1* neurons' inhibition results in increased thermogenesis and decreased vasoconstriction resulting in increased CBT. Conversely, excitation of the Adcyap1 thermoregulatory neurons results in hypothermia. Preoptic *Nts* neurons are GABAergic and project to preoptic thermoregulatory *Adcyap1* neurons and modulate their activity. Preoptic astrocytes express NtsR2 receptors and their activation modulates the release of glutamate from nearby synaptic terminals.

characteristics of the torpor state. A previous study has identified a population of preoptic *Adcyap1* neurons is involved in the entry in the torpor state (*Hrvatin et al., 2020*). Since in this study we found that preoptic *Adcyap1* neurons receive excitatory input from neurotensinergic neurons outside the preoptic area, it is possible that these projections are involved in the initiation of torpor state.

Experiments at the cellular level showed that locally applied CNO in slices from $Nts^{cre}$ $hM3D^{lox/lox}$ mice had a net excitatory effect in nearby $MPO^{Adcyap1}$ neurons. In addition to the actions observed during photostimulation of $MPO^{Nts;ChR2}$, namely increased sIPSCs frequency and an inward current, CNO also increased the frequency and amplitudes of sEPSCs recorded in $MPO^{Adcyap1}$ neurons. This may be due to activation of glutamatergic projections from brain regions that contain glutamatergic $Nts^{cre}$ $hM3D^{lox/lox}$ neurons. Such neurons have been identified in the posterior thalamus and in the ventrolateral periaqueductal gray (*Ma et al., 2019*; *Zhong et al., 2019*). It is also possible that some of the parabrachial nucleus neurons' glutamatergic projections (*Yang et al., 2020*) are also neurotensinergic. It is likely that the chemogenetic stimulation of MPO $Nts^{cre}$ $hM3D^{lox/lox}$ neurons and projections elevates the local Nts concentration to a level that activates the low affinity NtsR2. These receptors are expressed in MPO astrocytes and upon activation stimulate glutamate release from nearby synaptic terminals as observed when exogenous Nts is applied locally (*Tabarean, 2020*). A schematic model of the network and of the cellular mechanisms involved in the CBT modulation by neurotensinergic neurons is presented in *Figure 8*.

In summary, this study characterizes the cellular properties of $MPO^{Nts}$ neurons and their influence on CBT and provides insights in the cellular mechanisms at the MPO level that result in hypothermia.

## Materials and methods

**Key resources table**

| Reagent type (species) or resource | Designation | Source or reference | Identifiers | Additional information |
|---|---|---|---|---|
| Genetic reagent (*Mus musculus*) | B6;129-Nts*tm1(cre)Mgmj*/J; Slc32a1*tm1Lowl*; B6N;129-Tg(CAG-CHRM3*,-mCitrine)1Ute/J line | Jackson Laboratory Jackson Laboratory Jackson Laboratory | Cat. #:017525 RRID: IMSR_JAX:017525 Cat. #:012897 RRID:IMSR_JAX:012897 Cat. #:026220 RRID: IMSR_JAX:026220 | |
| Genetic reagent (AAV5) | AAV5-EF1a-double floxed-hChR2(H134R)-eYFP-WPRE-HGHpA; AAV5-EF1a-DIO-eYFP | Addgene Addgene | Cat. #:20298 Cat. #:27056 | |
| Sequence-based reagent | RNAscope Probe-Mm-Nts | ACDBio | Cat. #: 420441 | |
| Sequence-based reagent | RNAscope Probe-Mm-Slc32a1-C2 | ACDBio | Cat. #: 319191 C2 | |
| Sequence-based reagent | *Slc17a6F* external primer | This paper | PCR primers | CTGGATGGTCGTCAGTATTTTATG |
| Sequence-based reagent | *Slc17a6R* external primer | This paper | PCR primers | ATGAGAGTAGCCAACAACCAGAAG |
| Sequence-based reagent | *Slc17a6F* internal primer | This paper | PCR primers | GCAGGAGCTGGACTTTTTATTTAC |
| Sequence-based reagent | *Slc17a6R* internal primer | This paper | PCR primers | TAGTTGTTGAGAGAATTTGCTTGC |
| Sequence-based reagent | *NtsF* external primer | This paper | PCR primers | AGGCCCTACATTCTCAAGAG |
| Sequence-based reagent | *NtsR* external primer | This paper | PCR primers | CATTGTTCTGCTTTGGGTTA |
| Sequence-based reagent | *NtsF* internal primer | This paper | PCR primers | GGGGTTCCTACTACTACTGA |
| Sequence-based reagent | *NtsR* internal primer | This paper | PCR primers | CATCACATCCAATAAAGCAC |
| Sequence-based reagent | *Slc32a1F* PCR primers | This paper | PCR primers | GTCACGACAAACCCAAAGATCAC |
| Sequence-based reagent | *Slc32a1R* PCR primers | This paper | PCR primers | GTTGTTCCCTCATCATCTTCGCC |
| Sequence-based reagent | *Adcyap1F* PCR primers | This paper | PCR primers | CCTACCGCAAAGTCTTGGAC |
| Sequence-based reagent | *Adcyap1R* PCR primers | This paper | PCR primers | TTGACAGCCATTTGTTTTCG |
| Commercial assay or kit | Superscript III | Invitrogen | Cat. #:18080200 | |
| Chemical compound, drug | SR48692 | Tocris | Cat. #: 3721 | |
| Chemical compound, drug | NTRC 824 | Tocris | Cat. #:5438 | |
| Software, algorithm | pClamp10 | Molecular Devices | Version 10 | |
| Software, algorithm | MiniAnalysis software | Synaptosoft | Version 5 | |
| Software, algorithm | PolyScan2 | Mightex | Version 1 | |

## Animals

Experiments on animals were carried out in accordance with the National Institute of Health Guide for the care and use of Laboratory animals (1996 (7th ed.) Washington DC: National Research Council,

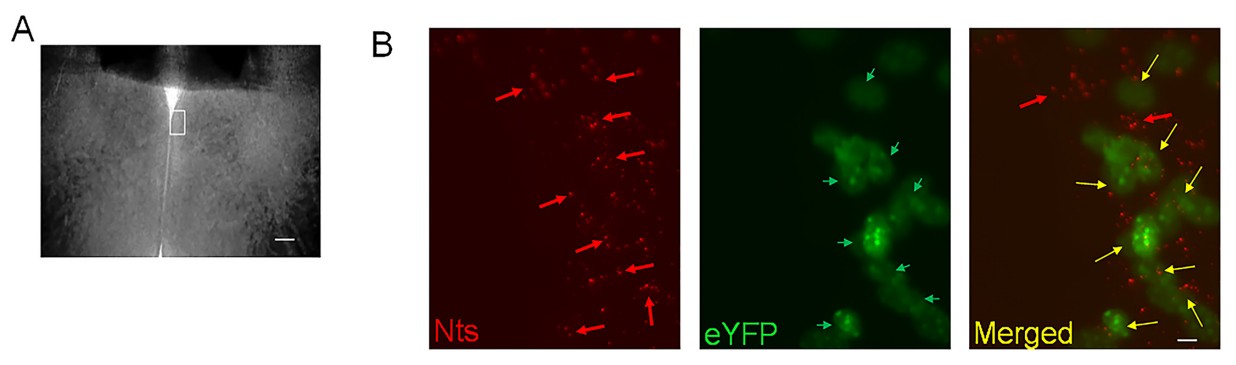

**Figure 9.** Transduction of MPO$^{Nts}$ neurons with ChR2-eYFP by injecting AAV-EF1a-double floxed-hChR2(H134R)-eYFP in Nts-cre mice. (**A**) Brightfield image of a MPO slice. The white rectangle represents the region imaged in (**B**). The scale bar represents 100 μm. (**B**) Representative images of Nts transcripts visualized using RNAscope (red, left panel), eYFP (green, middle) and their superimposed images (right) in a coronal slice from a MPO$^{Nts;ChR2}$ mouse. eYFP was visible in 8 out of 10 Nts positive cells. The scale bar represents 10 μm.

National Academies Press). The protocols were reviewed and approved by the Institutional Animal Care and Use Committee of the Scintillon Institute. The standards are set forth by the American Association for the Accreditation of Laboratory Animal Care (AAALAC) and in the Animal Welfare Act. The work was designed in such a way to minimize the number of animals used as well as their suffering.

The *Nts-cre* driver line (B6;129-Ntstm1(cre)Mgmj/J; stock no: 017525) (*St-Gelais et al., 2006*) was purchased from Jackson Laboratory (Bar Harbor, ME, USA). The *Nts*-cre driver line was crossed with the Slc32a1tm1Lowl (also referred to as VGATflox/flox) (Jackson Laboratory; stock no: 012897) to generate mice that have the GABA vesicular transporter Slc32a1 (solute carrier family 32 member 1), knocked-down in Nts neurons. Heterozygous double transgenic mice *Nts+/-;Slc32a1+/-* were crossed to obtain *Nts+/+;Slc32a1-/-*. These mice are referred to as *Nts$^{cre}$ Slc32a1$^{lox/lox}$*. We have also crossed the *Nts*-cre driver line with the B6N;129-Tg(CAG-CHRM3*,-mCitrine)1Ute/J line (Jackson Laboratory; stock no: 026220) to generate mice that express the excitatory designer receptor hM3D(Gq) in *Nts*-neurons in all brain regions. This new line was named *Nts$^{cre}$ hM3D$^{lox/lox}$*. In chemogenetics experiments, to activate hM3D(Gq), mice received an i.p. injection of CNO (0.7–0.8 mg/kg).

## Transgenic animals and AAV injections

The Nts-cre homozygous mice (4–6 months old) received bilateral stereotaxic injections (200 nL at a rate of 0.1 μL/min) of AAV-EF1a-double floxed-hChR2(H134R)-eYFP-WPRE-HGHpA (Addgene #20298; 2.1x10$^{13}$ virus molecules/ml) or AAV5-EF1a-DIO-eYFP (Addgene #27056; 1.3x10$^{13}$ virus molecules/ml) in the MPO to express the opsin Channelrhodopsin2 (ChR2) or eYFP (control) in MPO Nts neurons. We refer to these mice as MPO$^{Nts;ChR2}$ and MPO$^{Nts;eYFP}$, respectively. To ensure that only cre-expressing neurons were targeted we have carried out control injections of the two viral vectors in wild-type C57/Bl6 mice (3 mice for each) and confirmed that no fluorescent signal was detected in the brain of these mice. We have carried out RNAscope for Nts (see below) to determine the colocalization with eYFP (*Figure 9*) in three MPO$^{Nts;ChR2}$ mice. We have studied colocalization from six randomly chosen field of view from three slices from three different animals and found colocalization in 82% (355 out of 433) of the Nts positive neuros. Only 4 out of 359 eYFP neurons appeared to be Nts-negative (1.1%). Together with the detection of *Nts* transcripts in the labeled neurons (see Results section), we conclude that we have specifically targeted *Nts* neurons using these viral vectors.

Finally, we have performed bilateral stereotaxic injections (200 nL at a rate of 0.1 μL/min) of the AAV5-EF1a-double floxed-hChR2(H134R)-eYFP-WPRE-HGHpA (Addgene #20298; 2.1x10$^{13}$ virus molecules/ml) or AAV5-EF1a-DIO-eYFP (Addgene #27056; 1.3x10$^{13}$ virus molecules/ml) in the MPO of *Nts$^{cre}$ Slc32a1$^{lox/lox}$* to express ChR2 or eYFP, in MPO *Nts$^{cre}$ Slc32a1$^{lox/lox}$* neurons. We refer to these mice as MPO$^{Nts;Slc32a1-/-;ChR2}$ and MPO$^{Nts;Slc32a1-/-;eYFP}$, respectively.

At the end of in vivo experiments the brains were perfused, fixed and sliced to verify the site of AAV injection. The sections were examined under a dissection microscope. Only data from animals in which the MPO injection site was confirmed were included in this study.

All experiments were carried out in male mice. Key sets of experiments were also performed in female mice as specified in the text.

## Estrous cycle monitoring

Female mice have a 4–6 days estrous cycle that consists of four stages – proestrus, estrus, diestrus I/ metestrus, and diestrus/II. Mice were habituated to handling and vaginal smears prior to the behavioral experiments. Mice were swabbed daily before 9 a.m. and experiments started after 10 am. The stage of the cycle was determined using vaginal cytology. The smears were stained with hemalumeosin and analyzed microscopically. Experiments in females were carried out in diestrus days since during this stage the baseline CBT is less variable for each animal and among different animals. The days of diestrus are characterized by the presence of leukocytes and the absence of keratinized epithelial cells with no nuclei.

## Telemetry and MPO injections

For CBT measurements the mice (4–6 months old) were anesthetized with isoflurane (induction 3–5%, maintenance 1–1.5%) and radio telemetry devices (Anipill, BodyCap, Hérouville Saint-Clair, France) were surgically implanted into the peritoneal cavity. For MPO injections, mice were stereotaxically implanted with a bilateral guide cannula (27 Ga) as described in our previous studies (*Tabarean, 2021*). Coordinates for cannula implants in the MPO were: from Bregma: 0.05 mm, 0.35 mm, and −0.35 mm lateral, and ventral 4.75 mm (*Paxinos and Franklin, 2001*).The ambient temperature was maintained at ~28 ± 0.5°C in a 12:12 hr light-dark cycle-controlled room (lights on 8:00 am, ZT0). All substances injected were dissolved in sterile artificial cerebrospinal fluid (aCSF). For MPO injections, mice were placed in a stereotaxic frame and the injector (33 Ga) was lowered inside the cannula. A volume of 100 nL (rate 0.1 µL/min) was delivered using an injector connected to a microsyringe (0.25 µL). After injections the animal was returned to the home cage. Injections were carried out at 10 am local time, during the 'subjective light period'.

## Slice preparation

Coronal tissue slices containing the MPO were prepared from mice 4–6 months old $MPO^{Nts;ChR2}$, $MPO^{Nts;Slc32a1-/-;ChR2}$ or $Nts^{cre} hM3D^{lox/lox}$ mice. The slice preparation was as previously described (*Tabarean, 2021*). The slice used in our recordings corresponded to the sections located from 0.15 mm to −0.05 mm from Bregma in the mouse brain atlas (*Paxinos and Franklin, 2001*).

## Whole-cell patch-clamp recording

Whole-cell patch-clamp clamp was performed as described in our previous studies (*Tabarean, 2021*). The aCSF contained (in mM) the following: 130 NaCl, 3.5 KCl, 1.25 $NaH_2PO_4$, 24 $NaHCO_3$, 2 $CaCl_2$, 1 $MgSO_4$, and 10 glucose, osmolarity of 300–305 mOsm, equilibrated with 95% $O_2$ and 5% $CO_2$, pH 7.4. Other salts and agents were added to this saline. Whole-cell recordings were carried out using a $K^+$ pipette solution containing (in mM) 130 K-gluconate, 5 KCl, 10 HEPES, 2 $MgCl_2$, 0.5 EGTA, 2 ATP and 1 GTP (pH 7.3). The electrode resistance after back-filling was 2–4 MΩ. All voltages were corrected for the liquid junction potential (−13 mV). Data were acquired with a MultiClamp 700B amplifier (Molecular Devices, Sunnyvale, CA, USA) digitized using a Digidata 1550 interface and the Pclamp10.6 software package. The sampling rate for the continuous recordings of spontaneous activity was 50 kHz. The cell capacitance was determined and compensated using the Multiclamp Commander software.

The recording chamber was constantly perfused with extracellular solution (2–3 mL·min⁻¹). The antagonists were bath-applied. The bath temperature was maintained at 36–37°C by using an inline heater and a TC-344B temperature controller (Warner Instruments, Hamden, CT, USA).

Synaptic activity was quantified and analyzed statistically as described previously (*Tabarean, 2021*; *Lundius et al., 2010*). Synaptic events were detected and quantified (amplitude, kinetics, frequency) off-line using a peak detection program (Mini Analysis program, Synaptosoft, Decatur, NJ, USA). Events were detected from randomly selected recording stretches of 2 min before and during incubation with pharmacological agent. Cumulative distributions of the measured parameters (interevent interval, amplitude, rise time, time constant of decay) were compared statistically using with the Kolmogorov-Smirnov two-sample test (K-S test, p<0.05) using the Mini Analysis program. The averages for the measured parameters (frequency, amplitude, rise time, time constant of decay) for

each experiment were obtained using the Mini Analysis program. Event frequency was calculated by dividing the number of events by the duration (in seconds) of the analyzed recording stretch.

Spot illumination of MPO$^{Nts;ChR2}$ neurons was carried out using a Polygon300 illumination system (Mightex, Toronto, Canada) that allows the control of the size, shape, intensity and position of the illuminated spot as described previously. In a field of view there were 1–4 fluorescent neurons. The position and size of the spot were controlled using the PolyScan2 software (Mightex). The intensity of the light at the bottom of the recording chamber was measured using a photosensor (ThorLabs, Newton, NJ, USA) and ranged from 1.2 to 1.4 mW/mm$^{-2}$. The light pulses were 50ms long and delivered at 10 Hz for durations of 20–120 s.

## Optogenetic stimulation in vivo

A dual fiber-optic cannula (100 μm diameter, NA 0.22, 0.7 mm pitch; Doric Lens, Quebec, Canada) was implanted as described above for the guide cannula. Light pulses were applied using high-power photodiodes, a digital and analog I/O control module and Polygon 300 software (Mightex, Toronto, Canada) via dual fiber cables (Doric Lens, Quebec, Canada). The intensity of the light at the end of the optic cannula was measured using a fiber photodiode power sensor (ThorLabs, Newton, NJ, USA) and ranged from 3.9 to 4.3 mW/mm$^{-2}$. The light pulses were 50ms long and delivered at 10 Hz for 30 s periods followed by 10 s recovery breaks. The total duration of light stimulation was 60 min.

## Chemicals

Agonists and antagonists were purchased from Tocris (Ellisville, MO, USA). The other chemicals were from Sigma (Carlsbad, CA, USA).

## Cell harvesting, reverse transcription, and PCR

MPO neurons in slices were patch-clamped and then harvested into the patch pipette by applying negative pressure as previously described (*Tabarean, 2021*; *Lundius et al., 2010*). The content of the pipette was expelled in a PCR tube. dNTPs (0.5 mM), 50 ng random primers (Invitrogen) and H$_2$O were added to each cell to a volume of 16 μl. The samples were incubated at 65 °C for 5 min and then put on ice for 3 min. First strand buffer (Invitrogen), DTT (5 mM, Invitrogen), RNaseOUT (40 U, Invitrogen), and SuperScriptIII (200 U, Invitrogen) were added to each sample to a volume of 20 μl followed by incubation at room temperature for 5 min, at 50 °C for 50 min and then at 75 °C for 15 min. After reverse transcription samples were immediately put on ice. One μl of RNAse H was added to samples and kept at 37 °C for 20 min. PCR assays were carried out using the pairs of primers listed in *Supplementary file 1-table 1*. The PCR products were verified using Sanger sequencing. Two rounds of PCR, using nested primers, were used to detect *Nts* and *Slc17a6* transcripts. For *Adcyap1* and *Slc32a1* a second round of PCR was not necessary since it did not yield additional positives when compared with the first round. Two negative controls were routinely carried out. The first one was amplified from a harvested cell without reverse-transcription while the second was represented by a RT/PCR of the 'pipette tip' (e.g. obtained when a successful gigaseal was not achieved and the cytoplasm was not harvested by suction). Since all single MPO cells tested were negative to *Slc17a6* a second positive control was carried out. Ventromedial hypothalamic neurons, a predominantly glutamatergic population was tested. The positivity rate was 90% (9 of 10 neurons tested; *Figure 1—figure supplement 1B*).

## RNAscope assays

Tissue was processed using in situ hybridization to detect mRNA for *Nts*, *Slc32a1*. Forty-μm-thick coronal sections fixed in 4% PFA for 15 min at 4 °C, dehydrated in serial concentrations of ethanol (50–100%), and processed according to the protocol provided in the RNAscope kit (ACDbio Cat# 320293). Sections were hybridized with the following mixed probes; *Nts* (Mm-Nts, Cat. 420441) and *Slc32a1* (Mm-Slc32a1-C2, Cat. 319191-C2) for 2 hr at 40 °C.

## Endotoxin induced fever, heat and cold exposure tests

To induce a fever response we injected i.p. a dose of 0.1 mg/kg lipopolysaccharides (LPS) dissolved in 0.3 ml sterile saline. To study the CBT change to heat exposure the mice were placed, in their home cages, in a 37 °C incubator. Their CBT was monitored continuously by telemetry, and an animal was

removed from the incubator once it reached a CBT of 41.5 °C. For cold exposure tests the mice were placed in a cold room (4 °C) for 3 hr. All the experiments were started at 10 a.m. (ZT2).

## Data quantification and statistical analysis

The values reported are presented as mean ± standard deviation (SD). We used power analyses (https://www.biomath.info) with values from our data (means and SDs) to calculate that our study was powered to detect a 0.7 °C change in CBT with at least >80% reliability for all transgenic models used in this study. The normality of the samples was tested using the Shapiro-Wilk and Anderson-Darling tests. Statistical significance of the results pooled from two groups was assessed with t-tests or Mann-Whitney test using Prism9 (GraphPad Software). One-way analysis of variance (ANOVA, Kruskal-Wallis) with Tukey's post hoc test (p<0.05) was used for comparison of multiple groups. Cumulative distributions were compared with the Kolmogorov-Smirnov test (p<0.05). Data collected as time series were compared across time points by one-way ANOVA with repeated measures (p<0.05) (Prism4, GraphPad Software), followed by Mann Whitney U tests (p<0.05) for comparisons at each time point. The statistical value, the degrees of freedom and the p value are reported in the figure legends or, if data is not presented in a figure, the respective values are reported in the text. p Values for the results of Tukey's tests are presented in tables.

## Acknowledgements

This is manuscript number 1073 from the Scintillon Institute. This research was supported by the National Institutes of Health Grant NS094800 and the Norn Group Impetus grant 70051 (I.V.T.).

## Additional information

### Funding

| Funder | Grant reference number | Author |
| --- | --- | --- |
| National Institute of Neurological Disorders and Stroke | NS124844 | Iustin V Tabarean |
| National Institutes of Health | NS094800 | Iustin V Tabarean |
| Norn Group | Impetus grant 70051 | Iustin V Tabarean |

The funders had no role in study design, data collection and interpretation, or the decision to submit the work for publication.

### Author contributions

Iustin V Tabarean, Conceptualization, Formal analysis, Funding acquisition, Investigation, Writing – original draft, Writing – review and editing

### Author ORCIDs

Iustin V Tabarean ⬥ https://orcid.org/0000-0003-4615-1149

### Ethics

This study was performed in strict accordance with the recommendations in the Guide for the Care and Use of Laboratory Animals of the National Institutes of Health. All of the animals were handled according to approved institutional animal care and use committee (IACUC) protocols (2022-IT-001) of the Scintillon Institute. The protocol was approved by the Committee on the Ethics of Animal Experiments of the Scintillon Institute. All surgery was performed under isofluorane anesthesia, and every effort was made to minimize suffering.

Reviewer #1 (Public Review): https://doi.org/10.7554/eLife.98677.2.sa1
Reviewer #2 (Public Review): https://doi.org/10.7554/eLife.98677.2.sa2
Reviewer #3 (Public Review): https://doi.org/10.7554/eLife.98677.2.sa3

Author response https://doi.org/10.7554/eLife.98677.2.sa4

## Additional files

### Supplementary files
• MDAR checklist

• Supplementary file 1. Tables of PCR primers and of the statistical parameters obtained in the Tukey's test group comparisons.

### Data availability
*Figures 1–7*, *Figure 3—figure supplement 1*, *Figure 4—figure supplement 1* and *Figure 5—figure supplement 1* have source data files that include the numerical data used to generate the figures. *Figure 1—source data 2* and *Figure 2—source data 2* contain the original photographs of the gels presented in *Figures 1 and 2*.

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
