## [Editor Report · eLife assessment]

This is an **important** study to reveal local circuit mechanisms in the POA that control body temperature and also highlight how neurotransmitter GABA and neuropeptide NTS from the same neurons differentially modulate temperature. This study was carefully executed, providing **convincing** evidence for the conclusions in this paper. The findings have emphasized the importance of considering multiple diverse functions of the same neuron populations and will be of interest to neuroscientists working on central regulations of energy metabolism and temperature homeostasis.

---

## [Referee Report · Reviewer #1 (Public Review)]

Little is known about the local circuit mechanisms in the preoptic area (POA) that regulate body temperature. This carefully executed study investigates the role of GABAergic interneurons in the POA that express neurotensin (NTS). The principal finding is that GABA-release from these cells inhibits neighboring neurons, including warm-activated PACAP neurons, thereby promoting hyperthermia, whereas NTS released from these cells has the opposite effect, causing a delayed activation and hypothermia. This is shown through an elegant series of experiments that include slice recordings alongside matched in vivo functional manipulations. The roles of the two neurotransmitters are distinguished using a cell-type-specific knockout of Vgat as well as pharmacology to block GABA and NTS receptors. Overall, this is an excellent study that is noteworthy for revealing local circuit mechanisms in the POA that control body temperature and also for highlighting how amino acid neurotransmitters and neuropeptides released from the same cell can have opposing physiologic effects.

---

## [Referee Report · Reviewer #2 (Public Review)]

Summary:

The study has demonstrated how two neurotransmitters and neuromodulators from the same neurons can be regulated and utilized in thermoregulation.

The study utilized electrophysiological methods to examine the characteristics and thermoregulation of Neurotensin (Nts)-expressing neurons in the medial preoptic area (MPO). It was discovered that GABA and Nts may be co-released by neurons in MPO when communicating with their target neurons.

Strengths:

The study has leveraged optogenetic, chemogenetic, knockout, and pharmacological inhibitors to investigate the release process of Nts and GABA in controlling body temperature.

The findings are relevant to those interested in the various functions of specific neuron populations and their distinct regulatory mechanisms on neurotransmitter/neuromodulator activities

Weaknesses:

Key points for consideration include:

(1) The co-release of GABA and Nts is primarily inferred rather than directly proven. Providing more direct evidence for the release of GABA and the co-release of GABA and Nts would strengthen the argument. Further in vitro analysis could strengthen the conclusion regarding this co-releasing process.

(2) The differences between optogenetic and chemogenetic methods were not thoroughly investigated. A comparison of in vitro results and direct observation of release patterns could clarify the mechanisms of GABA release alone or in conjunction with Nts under different stimulation techniques.

(3) Neuronal transcripts were mainly identified through PCR, and alternative methods like single-cell sequencing could be explored.

(4) In Figure 6, the impact of GABA released from Nts neurons in MPO on CBT regulation appears to vary with ambient temperatures, requiring a more detailed explanation for better comprehension.

(5) The model should emphasize the key findings of the study.

---

## [Referee Report · Reviewer #3 (Public Review)]

Summary:

Understanding the central neural circuits regulating body temperature is critical for improving health outcomes in many disease conditions and in combating heat stress in an ever-warming environment. The authors present important and detailed new data that characterizes a specific population of POA neurons with a relationship to thermoregulation. The new insights provided in this manuscript are exactly what is needed to assemble a neural network model of the central thermoregulatory circuitry that will contribute significantly to our understanding of regulating the critical homeostatic variable of body temperature. These experiments were conducted with the expertise of an investigator with career-long experience in intracellular recordings from POA neurons. They were interpreted conservatively in the appropriate context of current literature.

The Introduction begins with "Homeotherms, including mammals, maintain core body temperature (CBT) within a narrow range", but this ignores the frequent hypothermic episodes of torpor that mice undergo triggered by cold exposure. Although the author does mention torpor briefly in the Discussion, since these experiments were carried out exclusively in mice, greater consideration (albeit speculative) of the potential for a role of MPO Nts neurons in torpor initiation or recovery is warranted. This is especially the case since some 'torpor neurons' have been characterized as PACAP-expressing and a population of PACAP neurons represent the target of MPO Nts neurons.

---

## [Author Response]

Author response:

**Public Reviews:**

**Reviewer #1 (Public Review):**
Little is known about the local circuit mechanisms in the preoptic area (POA) that regulate body temperature. This carefully executed study investigates the role of GABAergic interneurons in the POA that express neurotensin (NTS). The principal finding is that GABA-release from these cells inhibits neighboring neurons, including warm-activated PACAP neurons, thereby promoting hyperthermia, whereas NTS released from these cells has the opposite effect, causing a delayed activation and hypothermia. This is shown through an elegant series of experiments that include slice recordings alongside matched in vivo functional manipulations. The roles of the two neurotransmitters are distinguished using a cell-type-specific knockout of Vgat as well as pharmacology to block GABA and NTS receptors. Overall, this is an excellent study that is noteworthy for revealing local circuit mechanisms in the POA that control body temperature and also for highlighting how amino acid neurotransmitters and neuropeptides released from the same cell can have opposing physiologic effects. I have only minor suggestions for revision.
**Reviewer #2 (Public Review):**
Summary:The study has demonstrated how two neurotransmitters and neuromodulators from the same neurons can be regulated and utilized in thermoregulation.The study utilized electrophysiological methods to examine the characteristics and thermoregulation of Neurotensin (Nts)-expressing neurons in the medial preoptic area (MPO). It was discovered that GABA and Nts may be co-released by neurons in MPO when communicating with their target neurons.Strengths:The study has leveraged optogenetic, chemogenetic, knockout, and pharmacological inhibitors to investigate the release process of Nts and GABA in controlling body temperature.The findings are relevant to those interested in the various functions of specific neuron populations and their distinct regulatory mechanisms on neurotransmitter/neuromodulator activitiesWeaknesses:Key points for consideration include:(1) The co-release of GABA and Nts is primarily inferred rather than directly proven. Providing more direct evidence for the release of GABA and the co-release of GABA and Nts would strengthen the argument. Further in vitro analysis could strengthen the conclusion regarding this co-releasing process.

Measurement of Nts concentrations in various brain regions during thermoregulatory responses is part of a future study.

(2) The differences between optogenetic and chemogenetic methods were not thoroughly investigated. A comparison of in vitro results and direct observation of release patterns could clarify the mechanisms of GABA release alone or in conjunction with Nts under different stimulation techniques.

A comparison of chemogenetic and optogenetic stimulation methods is not within the scope of this study.

(3) Neuronal transcripts were mainly identified through PCR, and alternative methods like single-cell sequencing could be explored.

Single cell transcriptomics of preoptic neurotensinergic neurons will be part of a different study.

(4) In Figure 6, the impact of GABA released from Nts neurons in MPO on CBT regulation appears to vary with ambient temperatures, requiring a more detailed explanation for better comprehension.

The different possible roles of GABA in different thermoregulatory circumstances is discussed on lines 555-581.

(5) The model should emphasize the key findings of the study.

The model is presented in Fig 8.

**Reviewer #3 (Public Review):**
Summary:Understanding the central neural circuits regulating body temperature is critical for improving health outcomes in many disease conditions and in combating heat stress in an ever-warming environment. The authors present important and detailed new data that characterizes a specific population of POA neurons with a relationship to thermoregulation. The new insights provided in this manuscript are exactly what is needed to assemble a neural network model of the central thermoregulatory circuitry that will contribute significantly to our understanding of regulating the critical homeostatic variable of body temperature. These experiments were conducted with the expertise of an investigator with career-long experience in intracellular recordings from POA neurons. They were interpreted conservatively in the appropriate context of current literature.The Introduction begins with "Homeotherms, including mammals, maintain core body temperature (CBT) within a narrow range", but this ignores the frequent hypothermic episodes of torpor that mice undergo triggered by cold exposure. Although the author does mention torpor briefly in the Discussion, since these experiments were carried out exclusively in mice, greater consideration (albeit speculative) of the potential for a role of MPO Nts neurons in torpor initiation or recovery is warranted. This is especially the case since some 'torpor neurons' have been characterized as PACAP-expressing and a population of PACAP neurons represent the target of MPO Nts neurons.

Additional discussion of a possible role of neurotensinergic neurons in the initiation or recovery from torpor is included (lines 593-597).